


# The GGCMI Phase II experiment: global gridded crop model simulations under uniform changes in CO$_2$, temperature, water, and nitrogen levels (protocol version 1.0)

James Franke[1,2], Christoph Müller[3], Joshua Elliott[2,4], Alex C. Ruane[5], Jonas Jägermeyr[2,3,4,5], Juraj Balkovic[6,7], Philippe Ciais[8,9], Marie Dury[10], Pete Falloon[11], Christian Folberth[6], Louis François[10], Tobias Hank[12], Munir Hoffmann[13,22], R. Cesar Izaurralde[14,15], Ingrid Jacquemin[10], Curtis Jones[14], Nikolay Khabarov[6], Marian Koch[13], Michelle Li[2,16], Wenfeng Liu[8,17], Stefan Olin[18], Meridel Phillips[5,19], Thomas A. M. Pugh[20,21], Ashwan Reddy[14], Xuhui Wang[8,9], Karina Williams[11], Florian Zabel[12], and Elisabeth Moyer[1,2]

[1]Department of the Geophysical Sciences, University of Chicago, Chicago, IL, USA
[2]Center for Robust Decision-making on Climate and Energy Policy (RDCEP), University of Chicago, Chicago, IL, USA
[3]Potsdam Institute for Climate Impact Research, Member of the Leibniz Association, Potsdam, Germany
[4]Department of Computer Science, University of Chicago, Chicago, IL, USA
[5]NASA Goddard Institute for Space Studies, New York, NY, United States
[6]Ecosystem Services and Management Program, International Institute for Applied Systems Analysis, Laxenburg, Austria
[7]Department of Soil Science, Faculty of Natural Sciences, Comenius University in Bratislava, Bratislava, Slovak Republic
[8]Laboratoire des Sciences du Climat et de l'Environnement, CEA-CNRS-UVSQ, 91191 Gif-sur-Yvette, France
[9]Sino-French Institute of Earth System Sciences, College of Urban and Env. Sciences, Peking University, Beijing, China
[10]Unité de Modélisation du Climat et des Cycles Biogéochimiques, UR SPHERES, Institut d'Astrophysique et de Géophysique, University of Liège, Belgium
[11]Met Office Hadley Centre, Exeter, United Kingdom
[12]Department of Geography, Ludwig-Maximilians-Universität, Munich, Germany
[13]Georg-August-University Göttingen, Tropical Plant Production and Agricultural Systems Modeling, Göttingen, Germany
[14]Department of Geographical Sciences, University of Maryland, College Park, MD, USA
[15]Texas Agrilife Research and Extension, Texas A&M University, Temple, TX, USA
[16]Department of Statistics, University of Chicago, Chicago, IL, USA
[17]EAWAG, Swiss Federal Institute of Aquatic Science and Technology, Dübendorf, Switzerland
[18]Department of Physical Geography and Ecosystem Science, Lund University, Lund, Sweden
[19]Earth Institute Center for Climate Systems Research, Columbia University, New York, NY, USA
[20]School of Geography, Earth and Environmental Sciences, University of Birmingham, Birmingham, UK.
[21]Birmingham Institute of Forest Research, University of Birmingham, Birmingham, UK.
[22]Leibniz Centre for Agricultural Landscape Research (ZALF), D-15374 Müncheberg, Germany

**Correspondence:** Christoph Müller (cmueller@pik-potsdam.de)

**Abstract.** Concerns about food security under climate change motivate efforts to better understand future changes in crop yields. Process-based crop models, which represent plant physiological and soil processes, are necessary tools for this purpose since they allow representing future climate and management conditions not sampled in the historical record and new locations to which cultivation may shift. However, process-based crop models differ in many critical details, and their responses to different interacting factors remain only poorly understood. The Global Gridded Crop Model Intercomparison (GGCMI) Phase II experiment, an activity of the Agricultural Model Intercomparison and Improvement Project (AgMIP), is designed to provide





a systematic parameter sweep focused on climate change factors and their interaction with overall soil fertility, to allow both evaluating model behavior and emulating model responses in impact assessment tools. In this paper we describe the GGCMI Phase II experimental protocol and its simulation data archive. Twelve crop models simulate five crops with systematic uniform perturbations of historical climate, varying $CO_2$, temperature, water supply, and applied nitrogen ("CTWN") for rainfed and

irrigated agriculture, and a second set of simulations represents a type of adaptation by allowing the adjustment of growing season length. We present some crop yield results to illustrate general characteristics of the simulations and potential uses of the GGCMI Phase II archive. For example, modeled yields show robust decreases to warmer temperatures in almost all regions, with a nonlinear dependence that indicates yields in warmer baseline locations have greater temperature sensitivity. Inter-model uncertainty is qualitatively similar across all the four input dimensions, but is largest in high-latitude regions where crops may

be grown in the future.

## 1   Introduction

Understanding crop yield response to a changing climate is critically important, especially as the global food production system will face pressure from increased demand over the next century (Foley et al., 2005; Bodirsky et al., 2015). Climate-related reductions in supply could therefore have severe socioeconomic consequences (e.g. Stevanović et al., 2016; Wiebe

et al., 2015). Multiple studies using different crop or climate models concur in projecting sharp yield reductions on currently cultivated cropland under business-as-usual climate scenarios, although their yield projections show considerable spread (e.g. Rosenzweig et al., 2014; Schauberger et al., 2017; Porter et al. (IPCC), 2014, and references therein). Although forecasts of future yields reductions can be made with simple statistical models based on regressions in historical weather data, process-based models, which simulate the effect of temperature, water and nutrient availability, and atmospheric CO2 concentration on

the process of photosynthesis and the biology and phenology of individual crops, play a critical role in assessing the impacts of climate change.

Process-based models are necessary for understanding crop yields in novel conditions not included in historical data, including higher $CO_2$ levels, out-of-sample combinations of rainfall and temperature, cultivation in areas where crops are not currently grown, and differing management practices (e.g. Pugh et al., 2016; Roberts et al., 2017; Minoli et al., 2019). Process-

based models have therefore been widely used in studies on future food security (Wheeler and Von Braun, 2013; Elliott et al., 2014a; Frieler et al., 2017), options for climate mitigation (Müller et al., 2015) and adaptation (Challinor et al., 2018), and future sustainable development (Humpenöder et al., 2018; Jägermeyr et al., 2017). Process-based models also allow for the global gridded simulations needed for understanding the global dynamics of agricultural trade, including cultivation area changes and crop selection switching under climate change (Rosenzweig et al., 2018; Ruane et al., 2018) because global market mech-

anisms may strongly modulate climate change impacts (Stevanović et al., 2016; Hasegawa et al., 2018). Global crop model experiments are needed for systematic climate change assessments (Müller et al., 2017).

Modeling crop responses, however, continues to be challenging, as crop growth is a function of complex interactions between climate inputs, soil, and management practices (Boote et al., 2013; Rötter et al., 2011). Models tend to agree broadly in major





response patterns, including a reasonable representation of the spatial pattern in historical yields of major crops and projections of shifts in yield under future climate scenarios (e.g. Elliott et al., 2015; Müller et al., 2017). But process-based models still struggle with some important details, including reproducing historical year-to-year variability in many regions (e.g. Müller et al., 2017; Jägermeyr and Frieler, 2018), reproducing historical yields when driven by reanalysis weather (e.g. Glotter et al., 2014), and low sensitivity to extreme events (e.g. Glotter et al., 2015; Schewe et al., 2019). Long-term projections therefore retain considerable uncertainty (Wolf and Oijen, 2002; Jagtap and Jones, 2002; Iizumi et al., 2010; Angulo et al., 2013; Asseng et al., 2013, 2015).

Model intercomparison projects such as the Agricultural Model Intercomparison and Improvement Project (AgMIP, Rosenzweig et al., 2013) are crucial in quantifying uncertainties in model projections (Rosenzweig et al., 2014). Intercomparison projects have also been used to develop protocols for evaluating overall model performance (Elliott et al., 2015; Müller et al., 2017) and to assess the representation of individual physical mechanisms such as water stress and $CO_2$ fertilization (e.g. Schauberger et al., 2017). However, to date, few such projects have systematically sampled critical factors that may interact strongly in affecting crop yields. A number of modeling exercises in the last five years have begun to use systematic parameter sweeps in crop model evaluation and emulation (e.g. Ruane et al., 2014; Makowski et al., 2015; Pirttioja et al., 2015; Fronzek et al., 2018; Snyder et al., 2018; Ruiz-Ramos et al., 2018), but all involve limited sites and most also limited crops and scenarios.

The Global Gridded Crop Model Intercomparison (GGCMI) Phase II experiment is the first global gridded crop model intercomparison involving a systematic parameter sweep across critical interacting factors. GGCMI Phase II is an activity of AgMIP, and a continuation of a multi-model comparison exercise begun in 2014. The initial GGCMI Phase I (Elliott et al., 2015; Müller et al., 2017) compared harmonized yield simulations over the historical period, with primary goals of model evaluation and understanding sources of uncertainty (including model parameterization, weather inputs, and cultivation areas). See also Folberth et al. (2016); Porwollik et al. (2017) for more information. GGCMI Phase II compares simulations across a set of inputs with uniform perturbations to historical climatology, including $CO_2$, temperature, precipitation, and applied nitrogen (collectively referred to as "CTWN"), as well as adaptation to shifting growing seasons. The CTWN experiment is inspired by AgMIP's Coordinated Climate-Crop Modeling Project (C3MP Ruane et al., 2014; McDermid et al., 2015) and contributes to the AgMIP Coordinated Global and Regional Assessments (CGRA) (Ruane et al., 2018; Rosenzweig et al., 2018).

In this paper, we describe the GGCMI Phase II model experiments and present initial summary results. In the sections that follow, we describe the experimental goals and protocols; the different process-based models included in the intercomparison; the levels of participation by the individual models. We then provide an assessment of model fidelity based on observed yields at the country level, and show some selected examples of the simulation output dataset to illustrate model responses across the input dimensions.



## 2 Simulation objectives and protocol

### 2.1 Goals

The guiding scientific rationale of GGCMI Phase II is to provide a comprehensive, systematic evaluation of the response of process-based crop models to critical interacting factors, including $CO_2$, temperature, water, and applied nitrogen under two
contrasting assumptions on growing season adaptation (CTWN-A). The dataset is designed to allow researchers to:

- Enhance understanding of models' sensitivity to climate and nitrogen drivers.

- Study the interactions between climate variables and nitrogen inputs in driving modeled yield impacts.

- Characterize differences in crop responses to climate change across the Earth's climate regions.

- Provide a dataset that allows statistical emulation of crop model responses for downstream modelers.

- Explore the potential effects on future yield changes of adaptations in growing season length.

### 2.2 Modeling protocol

The GGCMI Phase I intercomparison was a relatively limited computational exercise, requiring yield simulations for 19 crops across a total of 310 model-years of historical scenarios, and had the participation of 14 modeling groups. The GGCMI Phase II protocol is substantially larger, involving over 1400 individual 30-year global scenarios, or over 42,000 model-years; 12
modeling groups nevertheless participated. To reduce the computational load, the GGCMI Phase II protocol reduces the number crops to 5 (maize, rice, soybean, spring wheat, and winter wheat). The reduced set of crops includes the three major global cereals and the major legume and accounts for over 50% of human calories in 2016: nearly 3.5 billion tons or 32% of total global crop production by weight (FAO, 2018). This set of major crops has the advantage of historical yield data globally available at sub-national scale (Ray et al., 2012; Iizumi et al., 2014), and has been frequently used in subsequent analyses (e.g.
Müller et al., 2017; Porwollik et al., 2017).

The Phase II protocol involves a suite of uniform perturbations from a historical climate timeseries. The baseline climate scenario for GGCMI Phase II is one of the weather products used in Phase I, daily climate inputs for 1980-2010 from the 0.5 degree NASA AgMERRA ("Agricultural"-modified Modern Era Retrospective analysis for Research and Applications) gridded re-analysis product. AgMERRA is specifically designed for agricultural modeling, with satellite-corrected precipitation (Ruane
et al., 2015). The experimental protocol consists of 9 levels for water supply perturbations, 7 for temperature, 4 for $CO_2$, and 3 for applied nitrogen, for a total of 756 simulations (Table 1), 672 for rainfed agriculture and an additional 84 for irrigated ($W_\infty$). In irrigated simulations, crops are assumed to have no water constraints, i.e. all crop water requirements are fulfilled regardless of local water supply limitations. Given that the irrigated scenario ($W_\infty$) is one element of the water supply levels, irrigated simulations use the same growing seasons and areas as all other simulations. All other water supply levels are implemented
as relative variations of precipitation. Values of climate variable perturbations are selected to represent reasonable ranges for





**Figure 1. Left panel:** Cultivated areas for maize, rice, and soybean are taken from the MIRCA2000 ("Monthly Irrigated and Rainfed Crop Areas around the year 2000") dataset (Portmann et al., 2010). Blue indicates grid cells with more that 20,000 hectares (10% of the equatorial grid cell) and gray contour shows gridcells with more that 10 hectares cultivated. Areas for winter and spring wheat areas are adapted from MIRCA2000 and two other sources; see text for details. For irrigated crops, see supplemental Figure S1. **Right panel:** Number of models providing simulations for each grid cell. All models provide the minimum areal coverage of the GGCMI Phase II protocol, but some provide extra coverage at high latitudes or in arid or otherwise unsuitable areas.





**Table 1.** GGCMI Phase II input parameter levels for each dimension. Temperature and precipitation values indicate the perturbations from the historical climatology. Irrigated ($W_\infty$) simulations assume the maximum beneficial levels of water. Bold font indicates the 'baseline' or historical level for each dimension. One model provided simulations at the T + 5 level.

| Input variable | Simulation input values | Unit |
|---|---|---|
| $CO_2$ (C) | **360**, 510, 660, 810 | ppm |
| Temperature (T) | -1, **0**, 1, 2, 3, 4, 6 | °C |
| Precipitation (W) | -50, -30, -20, -10, **0**, 10, 20, 30, (and $W_\infty$) | % |
| Applied nitrogen (N) | 10, 60, **200** | kg ha$^{-1}$ |
| Adaptation (A) | **A0: none**, A1: new cultivar to maintain original growing season length | - |

changes over the medium term (to 2100) under business-as-usual emissions. The resulting GGCMI Phase II dataset therefore captures the distribution of crop model responses over the range of potential future climate conditions.

While all perturbations are applied uniformly across the historical timeseries, they are applied in different ways. While precipitation perturbations are applied as fractional changes, temperature perturbations are applied as absolute offsets from the daily mean, minimum, and maximum temperature timeseries for each grid cell. $CO_2$ and nitrogen levels are specified as discrete values applied uniformly over all grid cells. The protocol samples over all possible permutations of individual perturbations. This choice means that $CO_2$ changes are applied independently of changes in climate variables, so that higher $CO_2$ is not associated with particular climate changes, e.g. higher temperatures.

Each model is run at 0.5 degree spatial resolution and covers all currently cultivated areas and much of the uncultivated land area. Figure 1, left, shows the present-day cultivated area of rainfed crops and Supplemental Figure S1-2 that for irrigated crops. Cultivated areas are provided by the MIRCA2000 (Monthly Irrigated and Rainfed Crop Area) data product (Portmann et al., 2010). Coverage extends considerably outside currently cultivated areas because cultivation will likely shift under climate change. To reduce the computational burden, however, the protocol requires simulation only over 80% of Earth land surface area. Areas are not simulated if they are assumed to remain non-arable even under an extreme climate change; these regions include Greenland, far-northern Canada, Siberia, Antarctica, the Gobi and Sahara Deserts, and Central Australia. The protocol also eliminates regions judged unsuitable for cropland for non-climatic reasons. Selection criterion involve a combination of soil suitability indices at 10 arc-minute resolution and excludes those 0.5 degree grid cells in which at least 90% of the area is masked as unsuitable according to any single index, and which do not contain any currently cultivated cropland. Soil suitability indices measure excess salt, oxygen availability, rooting conditions, toxicities, and workability, and are provided by the IIASA (International Institute for Applied Systems Analysis) Global Agro-Ecological Zone model (GAEZ, FAO/IIASA, 2011). The



procedure follows that proposed by Pugh et al. (2016). All modeling groups simulate the minimum required coverage, but some provide simulations that extend into masked zones, including e.g. the Sahara Desert and Central Australia (Figure 1, right).

### 2.3 Harmonization between models

The 12 models included in GGCMI Phase II are all process-based crop models that are widely used in impacts assessments (Table 3). Although some models share a common base (e.g. the LPJ family or the EPIC family of models), they have sub-sequently developed independently. Wherever possible, the GGCMI Phase II protocol harmonizes inputs, but differences in model structure mean that several key factors cannot be fully standardized across the experiment. These include soil treatment (which affects soil organic matter and carry-over effects of soil moisture across growing years) and baseline climate inputs.

While 10 of the 12 models participating in GGCMI Phase II use the AgMERRA historical daily climate data product, two models require sub-daily input data and thus use different baseline climate inputs: PROMET uses ERA-Interim reanalysis (Dee et al., 2011); JULES uses a bias-corrected version of ERA-Interim, the 3-hour WFDEI (WATCH-Forcing-Data-ERA-Interim) (Weedon et al., 2014), selecting WFDEI version with precipitation bias-corrected against the CRU TS3.101/TS3.21 precipitation totals (Harris et al., 2014). The data products show some differences (Figures S3-S4, which compare data products over currently cultivated areas for each crop). For example, for maize, ERA-Interim daily precipitation is biased high from that in AgMERRA by 7% (< 1 sigma), while mean daily precipitation in WFDEI is only 3% higher. Precipitation differences are largest in wheat areas, where ERA-Interim is substantially wetter (+60mm year$^{-1}$ or 10%). Temperatures for maize are very similar between data products, with ERA-Interim 0.45°C cooler and WFDEI 0.1°C warmer. Differences are largest for rice, with ERA-Interim 1°C cooler. These differences are relatively small compared to the perturbations tested in the protocol.

Planting dates and growing season lengths are standardized across models, following the procedure described in Elliott et al. (2015) for the *fullharm* setting. In contrast to GGCMI Phase I (Elliott et al., 2015), we here assume identical growing seasons for rainfed and irrigated scenarios, to allow for direct comparability of simulations along the W dimension, in which irrigation ($W_\infty$) is one element. (See Table 1.) While sowing dates are prescribed directly and held fixed in models, the length of the growing season is a product of crop phenology, which in turn is mostly driven by phenological parameters and temperature. Modelers are asked to adjust the phenological parameters so that growing season length of the baseline scenario (C=360, T=0, W=0) on average matches the harmonization target. Given that temperature varies between years, individual years can vary from the harmonization target. Growing seasons are harmonized across models but are crop- and location-specific. For example, at present maize is sown in March in Spain, in July in Indonesia, and in December in Namibia (Portmann et al., 2010). The one exception to the harmonization protocol described above involves CARAIB, which for technical reasons kept their own growing season specifications rather than tuning to standard lengths.

Because harvest dates are a function of climate parameters, simulations with the harmonized phenological parameters described above generally result in shorter growing seasons in future warmer scenarios. We denote these simulations as "A0" experiments, where 0 denotes "no adaptation". To account for potential adaptation in crop cultivars, the GGCMI Phase II protocol includes a second set of experiments, "A1", that assume that cultivars are modified to adjust to changes along the T dimension in the CTWN experiment. For these simulations, modelers adjust parameters to hold growing season length ap-





proximately constant across the different warming scenarios. (CARAIB simulations follow the same principle, fixing growing season length at their baseline levels.) The A1 simulations roughly capture the case in which adaptive crop cultivar choice ensures that crops reach maturity at roughly the same time as in the current temperature regime. This assumption is simplistic, and does not reflect realistic opportunities and limitations to adaptation (Vadez et al., 2012; Challinor et al., 2018), but provides
some insight into how crop modifications could alter projected impacts on yields and is sufficiently easy to implement in a large model intercomparison project as GGCMI.

Growing seasons for maize, rice, and soybean are taken from the SAGE (Center for Sustainability and the Global Environment, University of Wisconsin) crop calendar (Sacks et al., 2010), gap-filled with the MIRCA2000 crop calendar (Portmann et al., 2010) and, if no SAGE or MIRCA2000 data are available, with simulated LPJmL growing seasons (Waha et al., 2012)
and are identical to those used in GGCMI Phase I (Elliott et al., 2015). In GGCMI Phase II, we separately treat spring and winter wheat and so must define different growing seasons for each. As for the other crops, we use the SAGE crop calendar, which separately specifies spring and winter wheat, as the primary source for 69% of grid cells. In the remaining areas where no SAGE information is available, we turn to, in order of preference, the MIRCA2000 crop calendar (Portmann et al., 2010) and to simulated LPJmL growing seasons (Waha et al., 2012). These datasets each provide several options for wheat growing
season for each grid cell, but do not label them as spring or winter wheat. We assign a growing season to each wheat type for each location based on its baseline climate conditions. A growing season is assigned to winter wheat if all of the following hold, and to spring wheat otherwise:

- the monthly mean temperature is below freezing point (<0°C) at most for 5 months per year (i.e. winter is not too long)

- the coldest 3 months of a year are below 10°C (i.e. there is a winter)

- the season start date fits the criteria that:

  - if in the N. hemisphere, it is after the warmest *or* before the coldest month of the year (as winter is around the end/beginning of the calendar year)

  - if in the S. hemisphere, it is after the warmest *and* before the coldest month of the year (as winter is in the middle of the calendar year)

Nitrogen (N) application is standardized in timing across models. N fertilizer is applied in two doses, as is often the norm in actual practice, to reduce losses to the environment. In the GGCMI Phase II protocol, half of the total fertilizer input is applied at sowing and the other half on day 40 after sowing, for all crops except for winter wheat. For winter wheat, in practice the application date for the second N fertilizer application varies according to local temperature, because the length of winter dormancy can vary strongly. In the GGCMI Phase II protocol, the second fertilization date for winter wheat must lie at least
40 days after planting and – if not contradicting the distance to planting – no later than 50 days before maturity. If those limits permit, the second fertilization is set to the middle day of the first month after sowing that has average temperatures above 8°C.

All stresses in models are disabled other than those related to nitrogen, temperature, and water. For example, model responses to alkalinity, salinity, and non-nitrogen nutrients are all disabled. No other external N inputs are permitted – that is, there is no





atmospheric deposition of nitrogen – but some models allow additional release of plant-available nitrogen through mineralization in soils. In LPJmL, LPJ-GUESS and APSIM, soil mineralization is a part of model treatments of soil organic matter and cannot be disabled. Some additional differences in model structure mean that several key factors are not standardized across the experiment. For example, carry-over effects across growing years including residue management and soil moisture are treated

differently across models.

## 2.4 Output data products

All models in GGCMI Phase II provide 7 mandatory output variables if available (Table 2, bold). For each scenario, 0.5 degree grid cell and crop, models provide 30-year timeseries of annual crop yields in units of tons ha$^{-1}$ year$^{-1}$, as well as total aboveground biomass yield; the dates of planting, anthesis, and maturity; applied irrigation water in irrigated scenarios; and

total evapotranspirtation. (Note that several models do not output the anthesis date.) Besides these mandatory 7 data products, the protocol requests any or all of 18 optional additional output variables (Table 2, plain text). Participating modeling groups provided between 3 (PEPIC) and 18 (APSIM-UGOE) of these optional variables.

All output data is supplied as netCDF version 4 files, each containing values for one variable in a 30-year timeseries associated with a single scenario, for all grid cells. File names follow the naming conventions of GGCMI Phase I (Elliott et al.,

2015), which themselves are derived from those of ISIMIP (Frieler et al., 2017). File names are specified as

$[model]\_[climate]\_hist\_fullharm\_[variable]\_[crop]\_global\_annual\_[start-year]\_[end-year]\_[C]\_[T]\_[W]\_[N]\_[A].nc4$

Here $[model]$ is the crop model name; $[climate]$ is the original climate input dataset (typically AgMERRA); $[variable]$ is the output variable (of those in Table 2); $[crop]$ is the crop abbreviation ("mai" for maize, "ric" for rice, "soy" for soybean, "swh" for spring wheat, and "wwh" for winter wheat); and $[start-year]$ and $[end-year]$ specify the first and last years recorded

on file. $[C]$, $[T]$, $[W]$, $[N]$ and $[A]$ indicate the CTWN-A settings, each represented with the respective uppercase letter and the number indicating the level (e.g. "C360_T0_W0_N200" see Table 1). Except for the CTWN-A letters, the entire file name needs to be in small caps. All filenames include the identifiers $global$ and $annual$ to distinguish them as global, annual model output, following the updated ISIMIP file naming convention (Frieler et al., 2017).

Output data is provided on a regular geographic grid, identical for all models. Grid cell centers span latitudes -89.75 to

89.75° and longitudes from -179.75 to 179.75°. Missing values where no crop growth has been simulated are distinguished from crop failures: a crop failure is reported as zero yield but non-simulated areas (including ocean grid cells) have yields reported as "missing values" (defined as 1.e+20 in the netCDF files). Following NetCDF standards, latitude, longitude and time are included as separate variables in ascending order, with units "degrees north", "degrees east", and "growing seasons since 1980-01-01 00:00:00".

Following GGCMI Phase I standards, the first entry in each file describes the first complete cropping cycle simulated from the given climate input timeseries. In the AgMERRA timeseries used for GGCMI Phase II, the first year provided is 1980 but the date of the first entry can vary by crop and location. In the northern hemisphere, for summer crops like maize (sown in spring 1980 and harvested in fall 1980), the first harvest record would be of 1980, but for winter wheat (sown in fall 1980 and harvested in spring 1981) the first harvest record would be of 1981. Output files report the sequence of growing periods rather





**Table 2.** Output variables, naming convention, and units in the GGCMI Phase II protocol. Items in **bold** are the mandatory minimum requirements. Other variables are optionally provided depending on availability and participating modeling groups provided between 3 (PEPIC) and 18 (APSIM-UGOE) of these optional variables.

| Variable | variable name | units |
| --- | --- | --- |
| **Yield** | **yield_\<crop\>** | **t ha$^{-1}$ yr$^{-1}$ (dry matter)** |
| **Total above ground biomass yield** | **biom_\<crop\>** | **t ha$^{-1}$ yr$^{-1}$ (dry matter)** |
| **Actual planting date** | **plant-day_\<crop\>** | **day of year** |
| **Anthesis date** | **anth-day_\<crop\>** | **days from planting** |
| **Maturity date** | **maty-day_\<crop\>** | **days from planting** |
| **Applied irrigation water** | **pirrww_\<crop\>** | **mm yr$^{-1}$** |
| **Evapotranspiration (growing season sum)** | **etransp_\<crop\>** | **mm yr$^{-1}$ (W$_\infty$ scenarios only)** |
| Transpiration (growing season sum) | transp_\<crop\> | mm yr$^{-1}$ |
| Evaporation (growing season sum) | evap_\<crop\> | mm yr$^{-1}$ |
| Runoff (total growing season sum, subsurface + surface) | runoff_\<crop\> | mm yr$^{-1}$ |
| Total available soil moisture in root zone * | trzpah2o_\<crop\> | mm yr$^{-1}$ |
| Total root biomass | rootm_\<crop\> | t ha$^{-1}$ yr$^{-1}$ (dry matter) |
| Total Reactive Nitrogen (Nr) uptake (growing season sum) | tnrup_\<crop\> | kg ha$^{-1}$ yr$^{-1}$ |
| Total Nr inputs (growing season sum) | tnrin_\<crop\> | kg ha$^{-1}$ yr$^{-1}$ |
| Total Nr losses (growing season sum) | tnrloss_\<crop\> | kg ha$^{-1}$ yr$^{-1}$ |
| Gross primary production (GPP) | gpp_\<crop\> | gC m$^{-2}$ yr$^{-1}$ |
| Net primary production (NPP) | npp_\<crop\> | gC m$^{-2}$ yr$^{-1}$ |
| $CO_2$ response scaler on NPP | co2npp_\<crop\> | - {0..inf} |
| Water response scaler on NPP | h2onpp_\<crop\> | - {0..1} |
| Temperature response scaler on NPP | tnpp_\<crop\> | - {0..1} |
| Nr response scaler on NPP | nrnpp_\<crop\> | - {0..1} |
| Other nutrient response scaler on NPP | ornpp_\<crop\> | - {0..1} |
| $CO_2$ response scaler on transpiration | co2trans_\<crop\> | - {0..1} |
| Maximum stress response scaler | maxstress_\<crop\> | - {0..1} |
| Maximum Leaf Area Index (LAI) | laimax_\<crop\> | m$^2$ m$^{-2}$ |

* growing season sum, basis for computing average soil moisture



than calendar years. While there is generally one sowing event per calendar year (since simulations with harmonized growing seasons do not permit double-cropping), in some cases harvest events may skip or repeat within a calendar year. For example, because soybeans in North Carolina are typically harvested well into December, some calendar years may include no harvest (if it is not completed until after Dec. 31) or two harvests (one in January and one 11 months later in the following December).

## 3  Models contributing

The simulation output contributions of the 12 crop models to the GGCMI Phase II archive are described in Table 3. Not all modeling groups provided simulations for the full protocol described above. Given the substantial computational requirements, different participation tiers were specified to allow submission of smaller sub-sets of the full protocol. These subsets were designed as alternate samples across the 4 dimensions of the CTWN space, with *full* (12) and *low* (4) options for the C · N variables, and *full* (63), *reduced* (31), and *minimum* (9) options for T · W variables (described below). All participating modeling groups provided identical coverage of the CTWN parameter space for different crops, but most differed in CTWN coverage of A0 and A1 scenarios. Since the adaptation dimension was defined as a secondary priority for GGCMI Phase II, some models provided a more limited set of A1 scenarios. Of these, EPIC-IIASA, JULES, and ORCHIDEE-crop provided no A1 scenarios.

The different participation levels are defined by combining the CxN sets with the TxW sets:

- **full**: all 756 A0 simulations (all 12 CxN * all 63 TxW)

- **high**: 362 simulations (all 12 CxN combinations · *reduced* TxW set of 31 combinations)

- **mid**: 124 simulations (*low* 4 CxN combinations · *reduced* TxW set of 31 combinations)

- **low**: 36 simulations (*low* 4 CxN combinations · *minimum* TxW set of 9 combinations)

Of the 12 models submitting data, 6 followed the *full* protocol; these are marked with italic text in Table 3. However, note that two of these models (CARAIB and JULES) cannot represent nitrogen effects explicitly and so do not sample over the nitrogen dimension. Two models followed *high* with minor modifications (GEPIC adding an additional T level and PROMET omitting the intermediate N level). One model (PEPIC) followed *mid* but included an additional C level. Three models approximately followed *low* with APSIM-UGOE and EPIC-IIASA providing some additional TxW levels and ORCHIDEE-crop omitting some TxW combinations.

The combinations of perturbation values in the CxN and TxW parameter spaces used in the various participation levels are chosen to provide maximum coverage over plausible future values. For the CxN space, we specify:

- *full* as 12 pairs, with 4 C values (360, 660, 810 ppm) and 3 N (10, 60, 200 kg ha$^{-1}$ yr$^{-1}$)

- *low* as only 4 pairs: C360_N10, C360_N200, C660_N60, C810_N200

For the TxW space we specify:





**Table 3.** Models included in GGCMI Phase II and the number of CTWN-A simulations performed. The maximum number is 756 for A0 (no adaptation) experiments, and 648 for A1 (maintaining growing length) experiments, since T0 is not simulated under A1. "N-Dim." indicates whether the models are able to represent varying nitrogen levels. Each model provides the same set of CTWN simulations across all its modeled crops, but some models omit individual crops. (For example, APSIM-UGOE does not simulate winter wheat.)

| Model (Key Citations) | Maize | Soybean | Rice | Winter wheat | Spring wheat | N dim. | Sims per crop (A0 / A1) |
|---|---|---|---|---|---|---|---|
| **APSIM-UGOE**, Keating et al. (2003); Holzworth et al. (2014) | X | X | X | – | X | X | 44 / 36 |
| **CARAIB**, Dury et al. (2011); Pirttioja et al. (2015) | X | X | X | X | X | – | **252 / 216** |
| **EPIC-IIASA**, Balkovič et al. (2014) | X | X | X | X | X | X | 39 / 0 |
| **EPIC-TAMU**, Izaurralde et al. (2006) | X | X | X | X | X | X | **756 / 648** |
| **JULES**, Osborne et al. (2015); Williams and Falloon (2015); Williams et al. (2017) | X | X | X | – | X | – | **252 / 0** |
| **GEPIC**, Liu et al. (2007); Folberth et al. (2012) | X | X | X | X | X | X | 430 / 181 |
| **LPJ-GUESS**, Lindeskog et al. (2013); Olin et al. (2015) | X | – | – | X | X | X | **756 / 648** |
| **LPJmL**, von Bloh et al. (2018) | X | X | X | X | X | X | **756 / 648** |
| **ORCHIDEE-crop**, Wu et al. (2016) | X | – | X | X | – | X | 33 / 0 |
| **pDSSAT**, Elliott et al. (2014b); Jones et al. (2003) | X | X | X | X | X | X | **756 / 648** |
| **PEPIC**, Liu et al. (2016a, b) | X | X | X | X | X | X | 149 / 121 |
| **PROMET**, Hank et al. (2015); Mauser et al. (2015) | X | X | X | X | X | X | 261 / 232 |
| Totals | 12 | 10 | 11 | 10 | 11 | 10 | 5240 ǀ 3378 |

- *full* as all 7 T levels and 9W levels.

- *reduced* as 31 alternating combinations, with different Ws for even Ts than for odd Ts. For even Ts (i.e. T0,T2,T4,T6), we use W = -50,-20,0,+30 = 4·4 = 16 pairs. For odd Ts (i.e. T-1,T1,T3) , we use W = -30, -10, +10, +30, inf = 3·5 = 15 pairs.





– *minimum* as 9 combinations: T-1W-10, T0W10, T1W-30, T2W-50, T2W20, T3W30, T4W0, T4Winf, T6W-20

## 4 Results

To illustrate the properties of the GGCMI Phase II model simulations, we provide an evaluation of model performance by comparing model and historical yields, and show example results that demonstrate the spread of model responses to climate
and management inputs.

### 4.1 Evaluation of model performance

Evaluating the performance of crop models in the GGCMI Phase II archive is complicated by the artificial nature of the protocol: the settings in the CTWN-A experiment design do not reflect actual conditions in the real world. The protocol includes one scenario of near-historical climate inputs ($T_0$, $W_0$, $C_{360}$), but the prescribed uniform nitrogen application levels
do not reflect real-world fertilizer practices. Models also omit detailed calibrations to reflect the performance of historical cultivars.

We provide a partial evaluation of the models' skill in reproducing crop yield characteristics using the methodology of Müller et al. (2017), developed for GGCMI Phase I. Müller et al. (2017) evaluate how well model crop yield responses in a historical run capture real-world yield variations driven by year-to-year temperature and precipitation variations. Following this
approach, we compare yields in the GGCMI Phase II baseline simulations with detrended historical yields from the Food and Agriculture Organization of the United Nations (FAO, 2018) by calculating the Pearson product moment correlation coefficient over 26 years of yield. The procedure is sensitive to the detrending method and the area mask used to aggregate yields; we use a 5-year running mean removal and the MIRCA2000 cultivation area mask for aggregation. In some cases the model timeseries are shifted by one year to account for discrepancies in FAO or model year reporting. Because the GGCMI Phase II protocol
imposes fixed, uniform nitrogen application levels that are not realistic for individual countries, we evaluate control runs for each model at multiple N levels whenever possible. Nine of the GGCMI Phase II models (Table 3) provide historical runs for all three nitrogen levels (10, 60, and. 200 kg ha$^{-1}$ yr$^{-1}$).

As expected due to the unrealistic features described above, correlation coefficients for the GGCMI Phase II simulations are slightly lower than those found in the Phase I evaluation, but models show reasonable fidelity at capturing year-over-
year variation (Figure 2). For example, global correlation coefficients for maize in Phase I and Phase II are 0.89 and 0.74, respectively; for wheat 0.67 and 0.64, and for soybeans 0.64 and 0.59. (Compare to Müller et al. (2017) Figures 1–4 and 6.) Differences in fidelity between regions and crops exceed differences between models: that is, Figure 2 shows more color similarity in horizontal than vertical bars. For example, maize in the United States is consistently well-simulated while maize in Indonesia is problematic (mean Pearson correlation coefficients of 0.68 and 0.18, respectively). Note that in this methodology,
simulations of crops with low year-to-year variability such as irrigated rice and wheat will tend to score more poorly than those with higher variability. In some cases, especially in the developing world, low correlation coefficients may point to reporting problems in the FAO statistics and to real-world variability caused by variations in management rather than weather (Ray et al.,







**Figure 2.** Assessment of crop model performance in GGCMI Phase II, following the protocol of GGCMI Phase I (Müller et al., 2017). **Top:** example timeseries comparison between simulated crop yield and FAO country statistics (FAO, 2018) at the country level for two example high production countries: US maize, and rice in India, both for the 200 kg ha$^{-1}$ nitrogen application level. **Bottom:** heatmaps illustrating the Pearson $r$ correlation coefficient between the detrended simulated and observed country-level mean yields for the top 10 countries by production for each crop, of those countries with continuous FAO data over 1981-2010. We show separate comparisons for simulations with the three different nitrogen application levels, denoted 1, 2, 3 for 10, 60, and 200 kg N ha$^{-1}$, respectively. Left column shows correlation of ensemble mean yields with FAO data Because FAO does not distinguish between wheat types, we sum simulated spring and winter wheat for models that provide both (See Table 3.). Note that differences by region and crop are stronger than difference between models, e.g. horizontal bars are more similar in color than vertical bars. Countries are ordered alphabetically, not by production quantity.





2012; Müller et al., 2017). No single model consistently exhibits greater fidelity than others. Instead, each model shows near best-in-class performance for at least one location-crop combination. For example, pDSSAT is the best model for maize in the US, LPJmL and GEPIC are best in Germany, PROMET is best in Argentina, and PEPIC and LPJ-GUESS are best in France.

## 4.2 Model crop yield responses under CTWN forcing

Crop models in the GGCMI Phase II ensemble show broadly consistent responses to climate and management perturbations in most regions, with a strong negative impact of increased temperature in all but the coldest regions. Mapping the distribution of baseline yields and yield changes shows the geographic dependencies that underlie these results. Absolute yield potentials show strong spatial variation, with much of the Earth's surface area unsuitable for any of these crops (Figure 3, left). Crop yield changes under climate perturbations also show distinct geographic patterns (Figure 3, right, which shows fractional yield

differences between the T+4 scenario and the baseline scenario with historical climatology). In general, models agree most on yield response in regions where yield potentials are currently high and therefore where crops are currently grown. Models show robust decreases in yields at low latitudes, and highly uncertain ensemble mean increases at most high latitudes. Models show some increases in high mountain regions that are currently cold-limited.

Projections of strong yield growth at higher latitudes should be treated with caution, since the effects evident in Figure 3

are due in part to inaccuracies in model representations of present-day crop yields. For example, at latitudes north of 45°, the GGCMI Phase II models collectively suggest strong (but uncertain) growth in soybean yields under warmer conditions (Figure 3, g). However, model differences are greater in the baseline than future simulations, and greatest in currently-cultivated areas (Figure 4). Both the mean projected growth and the inter-model spread are driven by three models that show almost zero present-day potential soybean yields across the entire high-latitude region, even in locations where soybeans are currently

grown (Figure 4, left). PROMET, for example, involves a stronger response to cold than other models (e.g. LPJmL) with frost below -8 °C irreversibly killing non-winter crops and prolonged periods of below-optimum temperatures also leading to complete crop failure. Over the high-latitude regions simulated by both models, 52% of grid cells in PROMET report 0 yield in the present climate vs. 11% of cells in the T+4 scenario, leading to a strong yield gain in warmer future climates. In LPJmL outputs, the same high-latitude area is deemed suitable for cultivation even in baseline climate, with crop failure rates of 4%

and 5% in present and T+4 cases, so that projected yield changes are modest (Figure 4). These spurious low baseline yields result in very large fractional changes in the T+4 warming scenario, when all models agree that conditions become favorable for soybeans. Those models that most accurately reproduce present-day high-latitude soybean yields of 1-2 ton ha$^{-1}$ (Ray et al., 2012) in fact show a slight decrease in yield under a warming scenario (Figure 4, left). Apparent future yield increases in the multi-model mean are driven by the least realistic simulations.

The GGCMI Phase II exercise offers the opportunity to examine and characterize not just crop response to a single temperature change but nonlinearities in responses and interactions between factors. We illustrate a few of these relationships in Figures 5-6, choosing crops and factors whose effects are reasonably well understood. It is expected, for instance, that increases in precipitation should buffer the effects of warmer temperatures and that $CO_2$ increases should reduce damage to crops in scenarios where water is limited. Models generally confirm expected behavior but also provide insight into unforeseen



**Figure 3.** Illustration of the spatial patterns of baseline yields (left) and yield changes (right) in the GGCMI Phase II ensemble. Left column shows multi-model climatological(30 year) median yields for the baseline scenario, with white stippling indicating areas where these crops are not currently cultivated. Areas with less that 0.5 ton ha$^{-1}$ in the baseline are masked. Absence of cultivation aligns well with the lowest yield contour (0-2 ton ha$^{-1}$). Right column shows multi-model mean fractional yield changes in the T+4 °C scenario relative to the baseline scenario. Areas without stippling are those where models agree on changes: the multi-model mean fractional change exceeds the standard deviation of changes in individual models. Stippling indicates areas of low confidence ($\Delta < 1\sigma$). Some spatial structure in projected changes at high latitudes may be due to differences in model coverage; see Figure 1.



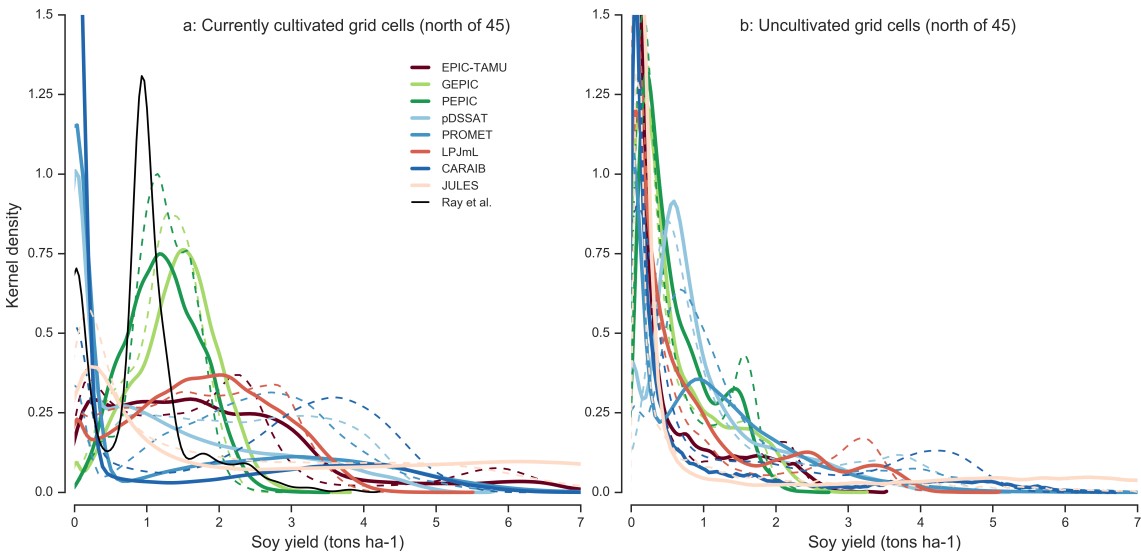

**Figure 4.** Model probability densities for soybean yields at latitudes north of $45°$ in historical and warming simulations in the A0 case. While 10 GGCMI Phase II models provide simulations (Table 3); we show 8 representative models for clarity. Probability density functions are estimated separately for locations with some current cultivation (left, approximately 2500 grid cells, unweighted by cultivated area) and for uncultivated locations (right, approximately 1500 grid cells), for baseline historical (solid) and T+4 ($°C$) (dashed) simulations. Black line in left panel shows actual yields from 1997-2003 derived from Ray et al. (2012). For historical simulations, models agree on low potential yields in currently uncultivated areas (right) but disagree widely on yields in currently cultivated areas (left). Color code groups models into those with realistic yield distributions peaking at 1-2 ton ha$^{-1}$ (green), those with flatter distributions extending to unrealistically high values (red), and those with predominantly zero yields (blue). "Green" models show slight decreases under T+4 warming, "red" models moderate increases, and "blue" models large increases.

interactions. To show geographic effects, we divide model responses in Figures 5-6 by the primary Köppen-Geiger climate regions (Rubel and Kottek, 2010), showing the yield changes across all simulated grid cells in each region. In each panel we examine relationships between two factors, showing yield response against one for several scenarios of the other, in box plots that show the inter-model spread. The responses highlighted here are qualitatively similar across all crops included in this study

5 (Supplemental Figures S5 - S8).

For all crops, warming scenarios with precipitation held constant produce yield decreases in most regions. These impacts are robust for even moderate climate perturbations. For rainfed maize, even a 1°C temperature increase with other factors held constant produces a median regional decline in potential yield that exceeds the variance across models, in all but the "cold-continental" regions (Figure 5a). The remaining areas ("warm temperate", "equatorial", and "arid") account for nearly

10 three-quarters of global maize production. In the high-latitude "cold-continental" region, potential yield changes are positive but highly uncertain, for the reasons discussed previously; uncertainties are larger even for maize than for soybeans. (Compare Figures 5a and 5b.) Temperature effects are somewhat nonlinear, with the largest impacts for maize in the warm "tropical"



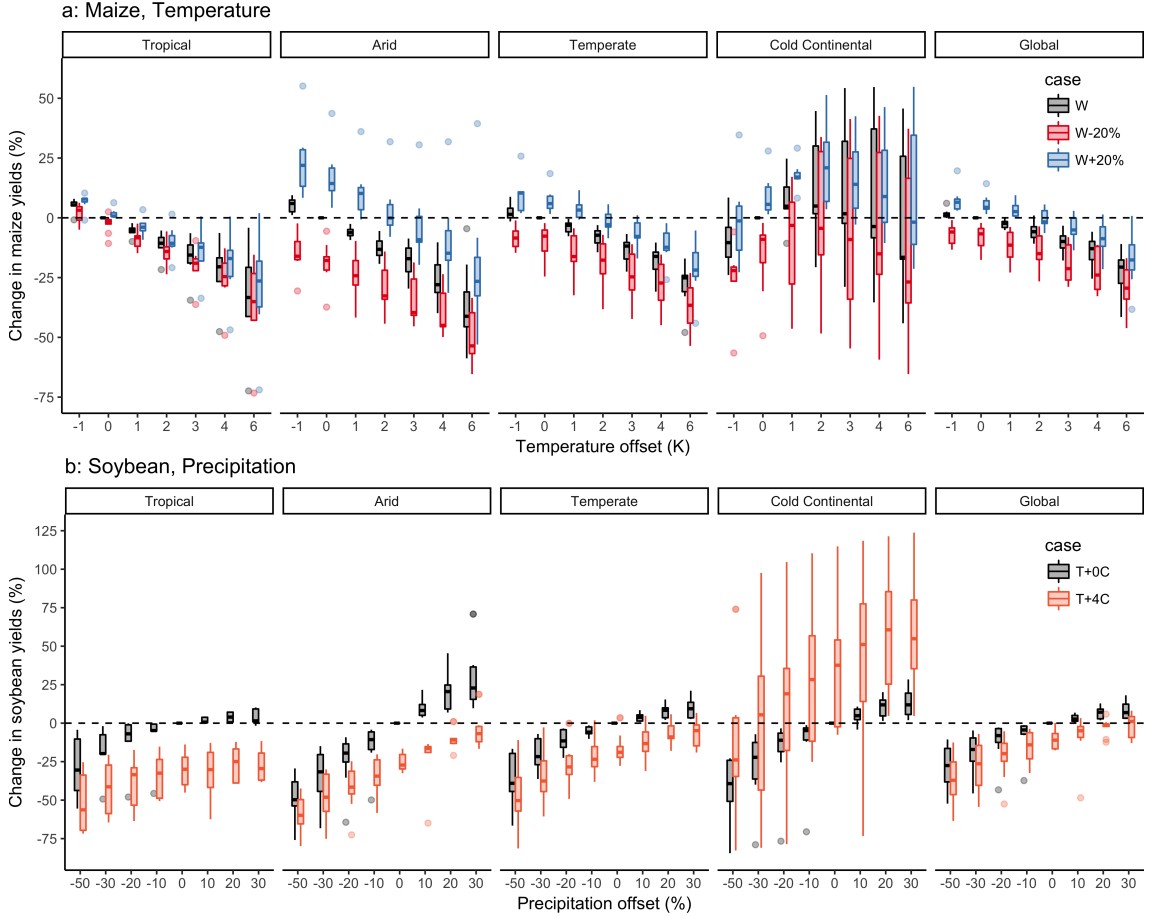

**Figure 5.** Illustration of the distribution of regional yield changes across the multi-model ensemble, split by Köppen-Geiger climate regions, and with global response in rightmost panel. Y-axis is the fractional change in the regional average climatological (30-year mean) potential yield relative to the baseline. Box-and-whiskers plots show distribution across models, with median marked; edges are first and third quartiles and whiskers extend to 1.5·IQR. Figure shows all all simulated grid cells for each model; see Supplemental Figure S10-S13 for only currently-cultivated land. We highlight responses to individual factors; note that results are not directly comparable to simulations of realistic projected climate scenarios with identical global mean changes. Models generally agree outside high-latitude regions, with projected changes exceeding inter-model variance. **Top:** Response of rainfed maize to applied uniform temperature perturbations, for three discrete precipitation perturbation levels ( -20%, 0%, and +20%), with $CO_2$ and nitrogen held constant at baseline values (360 pmm and 200 kg ha$^{-1}$ yr$^{-1}$). Outliers in the tropics (strong negative impact of higher T) are the pDSSAT model; outliers in the arid region (strong positive impact of higher P) are JULES. **Bottom:** Response of rainfed soybeans to applied uniform precipitation perturbations, for two discrete temperature levels. Cases with reduced precipitation show greater inter-model spread than those with increased precipitation. At very large precipitation increases, yield changes level out: benefits saturate once water availability is no longer limiting. Precipitation changes are more important in the arid region, as expected. Note the large uncertainty in the cold continental region, also illustrated in Figures 3 and 4.



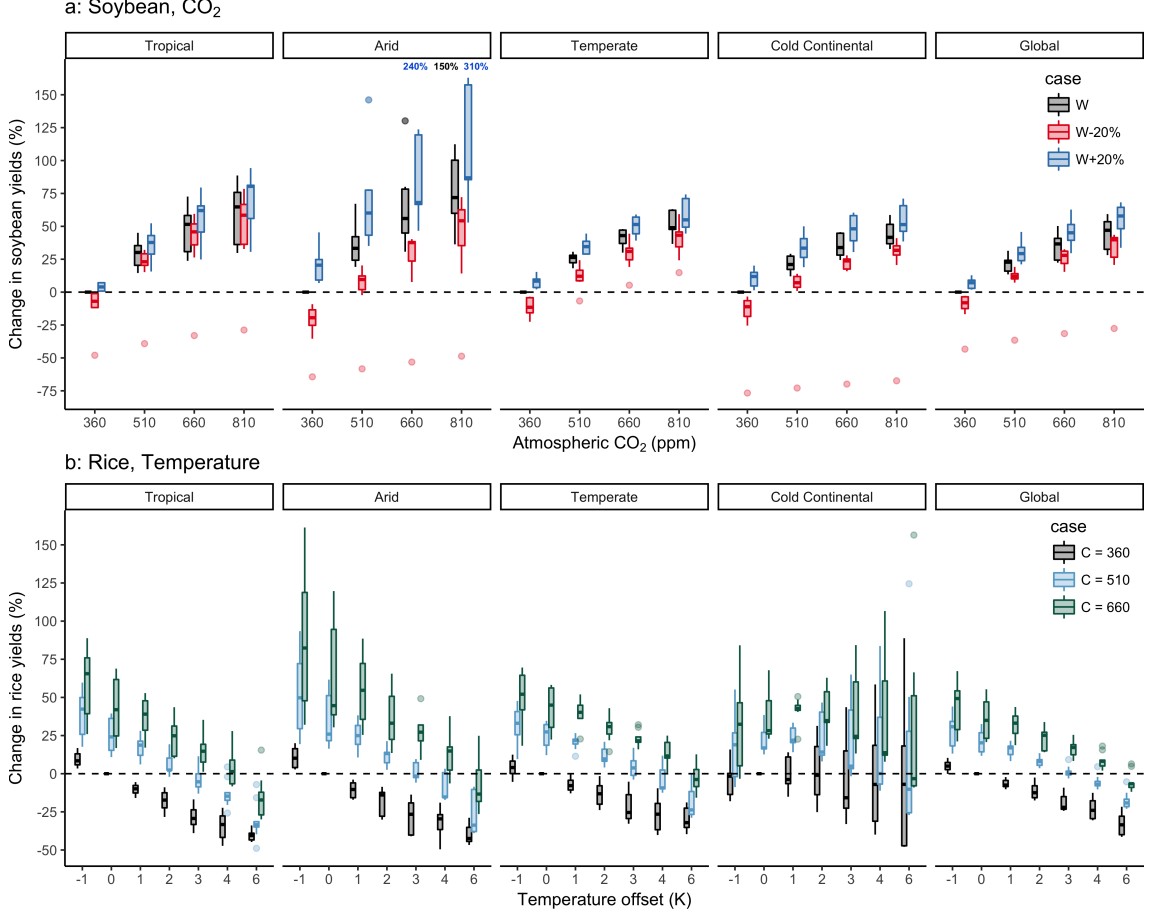

**Figure 6.** Illustration of the distribution of regional yield changes across the multi-model ensemble, here for soybeans and rice for the A0 case. Conventions as in Figure 5. **Top:** Response of rainfed soybeans to atmospheric $CO_2$, for three discrete precipitation perturbation levels with temperature and nitrogen held constant at baseline values. Low outliers are the EPIC-TAMU model and the high outliers in the Arid region are the JULES model. Reduced precipitation tends to steepen the $CO_2$ response and increased precipitation tends to flatten it, as expected. Reduced precipitation tends to increase the inter model spread, especially at the highest $CO_2$ levels. **Bottom:** Response of irrigated rice for three discrete $CO_2$ levels, with nitrogen and precipitation held constant. $CO_2$ does not change the nature of temperature response respective to baseline as the slopes at each $CO_2$ level are relatively constant.

region. (For soybeans, temperature effects are more complex; see Supplemental Figure S5.) Precipitation effects on rainfed crops are more strongly nonlinear. The curvature of the precipitation response can be seen by eye in Figure 5b: soybean yields are strongly negatively impacted by reduced rainfall, peak under increased precipitation of 20%, and actually decline at higher precipitation levels.

5    As expected, precipitation and temperature effects interact, with increases in precipitation buffering yield responses to temperature. Increased rainfall mitigates the negative impacts of warmer temperatures caused by increased evapotranspiration (e.g.





Allen et al., 1998). For maize, the effect is relatively modest outside the "arid" regions (Figure 5a). Globally, a $4°C$ temperature rise with no change in precipitation results in median loss of $\sim13\%$ of rainfed maize, with all models showing a negative response. With a 20% increase in precipitation, the median yield loss is $\sim8\%$. For soybeans, the equivalent values are $\sim11\%$ and 1%, respectively. Decreased rainfall, on the other hand, amplifies yield losses and also increases inter-model variance. That is,

models agree that the response to decreased water availability is negative in sign but disagree on its magnitude. Outside of arid regions, the interaction effect itself shows little nonlinearity (i.e. response slopes in Figures 5a and 5b are roughly parallel). As expected, irrigated crops are more resilient to temperature increases in all regions, especially so where water is already limiting (other than winter wheat, Supplemental Figure S9).

Increased $CO_2$ boosts yields overall through the well-known $CO_2$ fertilization effect (Figure 6). The effect is strongest for

the C3 crops (wheat, soybeans, and rice), while maize, a C4 grass, has a comparatively muted response. We show irrigated rice and rainfed soy in Figure 6 as representative C3 crops. The effect of $CO_2$ on yields is nonlinear, as expected, with significant benefit from small increases but with effects plateauing at higher concentrations (Figure 6). $CO_2$ and temperature effects show minimal interaction. This effect is seen in Figure 6a, which shows nearly parallel response slopes at different $CO_2$ levels. That is, $CO_2$ fertilization does little to change the nature of the temperature response. On the other hand, $CO_2$ and precipitation

effects interact strongly, as expected since higher $CO_2$ levels allow reduced stomatal conductance and evapotranspiration losses, mitigating the effect of reduced rainfall. This interaction is seen in Figure 6b as smaller yield losses from reduced rainfall when $CO_2$ levels are higher. For example, for soy, raising $CO_2$ to 510 ppm actually outweighs the multi-model median damages caused by a 20% precipitation reduction in all climate regions. All crops show similar behavior, but note that model uncertainties for wheat are substantially higher than those for other crops. (Compare Figure 6a for soy and Supplemental Figure

S7 for wheat).

## 5   Discussion and Conclusions

The GGCMI Phase II experiment provides a database designed to allow detailed study of crop yields in process-based models under climate change. While previous crop model intercomparison projects in the climate change context have focused on simulations along realistic projected climate scenarios (e.g Rosenzweig et al., 2014), the use of systematic input parameter

variations in GGCMI Phase II (with up to 756 scenarios) allows not only comparing yield sensitivities to changing climate and management inputs but also evaluating the complex interactions between important driving factors: $CO_2$, temperature, water supply, and applied Nitrogen. The global extent of the experiment also allows identifying geographic shifts in high potential yield locations. With 12 participating models and 31 simulation years per scenario the complete database constitutes over 150,000 years of gridded global yield simulation output for each crop.

Preliminary results shown here highlight some of the insights facilitated by the simulation exercise and lend confidence in the models. In validation tests of simulations of the historical scenario, year-over-year correlations in modeled and actual country-level yields are similar to those of GGCMI Phase I. In simulations on scenarios with perturbed climate and management factors, models generally agree on changes driven outside the high latitudes, with changes at nearly all perturbation levels exceeding





the inter-model spread. (At high latitudes, differences are often driven by differences in model treatment of crop response to current cold conditions.) In simulations with multiple perturbations, interactions between major yield drivers (temperature and precipitation in Figure 5, precipitation and $CO_2$ in Figure 6) generally follow expectations and produce physically reasonable responses in crop yields.

Users should however be aware of some limitations of the experiment that affect its potential applications. First, absolute model yield values in the historical scenario will generally not match observed yields. In order to match current yields, process-based models must generally be re-tuned to account for the constant evolution of crop cultivar genetics and management practice (e.g. Jones et al., 2017). The historical scenario also includes no trend in $CO_2$, and scenarios assume unrealistic globally uniform nitrogen application levels (Elliott et al., 2015). GGCMI Phase II is intended as a study of model-projected

changes under broad climate change, which may not be as sensitive to the adjustments needed to reproduce present-day yields. Note however that the models used in this exercise cannot simulate some potential climate-related yield changes: those due to factors such as pests, diseases, and weeds.

The second major caveat is that no individual GGCMI Phase II simulation is itself a realistic future yield projection. The uniform applied offsets in temperature and precipitation sample over potential changes, but projections of climate change involve spatially heterogeneous warming and precipitation changes. GGCMI Phase II simulation results can be used for impacts

projection, but only with the construction of an emulator of crop yield response to climatological changes, that is then driven by a realistic climate projection. However, note that the experiment does not sample over potential changes in the higher-order moments in the distributions in temperature and precipitation.

We expect that the GGCMI Phase II simulations will yield multiple insights in future studies. Potential applications include, as mentioned, the construction of emulators and yield response surface, as well as studies of issues such as the benefits of

adaptation, interactions between the CTWN factors affecting yield, changes in nitrogen use efficiency, geographic shifts in regional production, investigation of core sensitivities to CTWN/A by region and farm system, identification of hotspots of crop system vulnerability, rapid assessment of new climate projections, and many others. In general, the development of multi-model ensembles involving systematic parameters sweeps has large promise both for increasing understanding of potential

future crop responses and for improving process-based crop models.

*Code and data availability.* The simulation outputs of the mandatory 7 output variables (Table 2) are available on zenodo.org. See Appendix A1 for data DOIs. All other simulation output variables are available upon request to the corresponding author. The scripts for generating the spring wheat and winter wheat growing seasons and second fertilizer dates and the quality screening script is available at https://github.com/RDCEP/ggcmi/blob/phase2/. All input data are available via globus.org (registration required, free of charge):

Minimum cropland mask is available at https://app.globus.org/file-manager?origin_id=e4c16e81-6d04-11e5-ba46-22000b92c6ec& origin_path=%2FAgMIP.input%2Fother.inputs%2Fphase2.masks%2F choose the file boolean_cropmask_ggcmi_phase2.nc4 Growing period data for wheat is now divided up into winter and spring wheat, available at https://app.globus.org/file-manager?origin_id= e4c16e81-6d04-11e5-ba46-22000b92c6ec&origin_path=%2FAgMIP.input%2Fother.inputs%2FAGMIP_GROWING_SEASON.HARM. version2.0%2F whereas all other growing season data (maize, rice, soybean) are the same as in Phase I (version 1.25), available





at https://app.globus.org/file-manager?origin_id=e4c16e81-6d04-11e5-ba46-22000b92c6ec&origin_path=%2FAgMIP.input%2Fother.
inputs%2FAGMIP_GROWING_SEASON.HARM.version1.25%2F

# Appendix A

## A1 Data Access

5 Simulation yield output datasets can be found at the DOIs located in table A1. Data are published in crop- and GGCM-specific packages, in order to break down the overall data amount into manageable packages (<50GB per archive).

**Table A1.** DOI's for model data outputs. All model output data can be found at https://doi.org/10.5281/zenodo/XX. Where XX is the value listed in the table.

| Model | Maize | Soybean | Rice | Winter wheat | Spring wheat |
|-------|-------|---------|------|--------------|--------------|
| **APSIM-UGOE** | 2582531 | 2582535 | 2582533 | 2582537 | 2582539 |
| **CARAIB** | 2582522 | 2582508 | 2582504 | 2582516 | 2582499 |
| **EPIC-IIASA** | 2582453 | 2582461 | 2582457 | 2582463 | 2582465 |
| **EPIC-TAMU** | 2582349 | 2582367 | 2582352 | 2582392 | 2582418 |
| **JULES** | 2582543 | 2582547 | 2582545 | – | 2582551 |
| **GEPIC** | 2582247 | 2582258 | 2582251 | 2582260 | 2582263 |
| **LPJ-GUESS** | 2581625 | – | – | 2581638 | 2581640 |
| **LPJmL** | 2581356 | 2581498 | 2581436 | 2581565 | 2581606 |
| **ORCHIDEE-crop** | 2582441 | – | 2582445 | 2582449 | – |
| **pDSSAT** | 2582111 | 2582147 | 2582127 | 2582163 | 2582178 |
| **PEPIC** | 2582341 | 2582433 | 2582343 | 2582439 | 2582455 |
| **PROMET** | 2582467 | 2582488 | 2582479 | 2582490 | 2582492 |

*Author contributions.* J.E., C.M, and A.R. designed the research. C.M., J.J., J.B., P.C., M.D., P.F., C.F., L.F., M.H., C.I., I.J., C.J., N.K., M.K., W.L., S.O., M.P., T.P., A.R., X.W., K.W., and F.Z. performed the simulations. J.F., J.J., C.M., and E.M. performed the analysis and J.F., E.M., and C.M. prepared the manuscript.



*Competing interests.* The authors declare no competing interests.

*Acknowledgements.* This research was performed as part of the Center for Robust Decision-Making on Climate and Energy Policy (RDCEP) at the University of Chicago, and was supported through a variety of sources. RDCEP is funded by NSF grant #SES-1463644 through the Decision Making Under Uncertainty program. J.F. was supported by the NSF NRT program, grant #DGE-1735359. C.M. was supported by
5    the MACMIT project (01LN1317A) funded through the German Federal Ministry of Education and Research (BMBF). C.F. was supported by the European Research Council Synergy grant #ERC-2013-SynG-610028 Imbalance-P. P.F. and K.W. were supported by the Newton Fund through the Met Office Climate Science for Service Partnership Brazil (CSSP Brazil). K.W. was supported by the IMPREX research project supported by the European Commission under the Horizon 2020 Framework programme, grant #641811. S.O. acknowledges support from the Swedish strong research areas BECC and MERGE together with support from LUCCI (Lund University Centre for studies of Carbon
10    Cycle and Climate Interactions). R.C.I. acknowledges support from the Texas Agrilife Research and 634 Extension, Texas A & M University. This is paper number 35 of the Birmingham Institute of Forest Research. Computing resources were provided by the University of Chicago Research Computing Center (RCC).



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
