# Peer review of "The GGCMI Phase II experiment: global gridded crop model simulations under uniform changes in CO2, temperature, water, and nitrogen levels (protocol version 1.0)"

_Geoscientific Model Development, 2019_

## Referee Comment (RC1) · Anonymous Referee #1 · 10 Dec 2019

The manuscript by Franke et al., details the experimental design for the Phase II GGCMI crop model comparison. The goal is to provide a set of simulations to synchronize a variety of crop models and compare the responses from perturbations of temperature, precipitation, CO2, and nitrogen fertilizer. The result is a dataset of thousands of simulations that can be used to emulate statistical crop model response under varying inputs of climate change. The authors provide some analysis of the dataset, providing examples of non-linear behavior under multiple variable perturbations between temperature, precipitation, and CO2. Furthermore, the authors provide access

to other users for additional analysis. The manuscript is well written, the message is clearly defined, with a logical flow throughout, and void of technical errors. The authors did a good job detailing some of the more complex features of their study.

My main concern with this manuscript is I find the approach toward the perturbation experiments somewhat unrealistic. I understand the difficulty in generating simulations across different models in a way that is uniform, and I find the large number of simulations included in the dataset impressive but having such a large set of parameters for the simulations makes interpreting the output difficult and negates the heterogeneous (in space and time) behavior of climate. Wouldn't it be easier to use CMIP output to drive simulations which could reduce the number of model runs? Perhaps the authors could provide more discussion on this choice. Also, since these are offline runs, they don't include feedbacks between the atmosphere and land (e.g. irrigation feedbacks to temperature), which are important.

I did not find the A1 simulations discussed anywhere. They seem to be included in the methods section but are not included in the analysis. Perhaps they should be omitted. Similarly, the nitrogen simulations are also missing from the analysis (except for the correlation with observations).

General Comments:

P. 7, Section 2.3: The 12 models included in the study are very different types of models. I know this was discussed in the original paper describing protocol I, but it should also be noted here. How did the model differences inform the experimental design (or limit the scope of the study)?

P. 9, L. 10: If some models don't output the anthesis date, why is it considered mandatory?

P. 15, L. 6: Is the negative impact on yield from increasing temperature due to shorter growing seasons or from actual heat damage to the crop?

P. 15, L 11-13: The change in yields at different latitudes is unrealistic because of the design of the experiment. Simply increasing temperature uniformly and not accounting for the seasonal differences in temperature change (i.e., stronger winter increase in temperature and weak or no summer increase) results in an unrealistic "warming" during the growing season that might not exist. This is also the probable cause of the increase in yield from the least realistic simulations (Pl. 15, L 28-29).

---

## Referee Comment (RC2) · Anonymous Referee #2 · 4 Jan 2020

The manuscript by Franke et al. documented a new AgMIP GGCMI effort on simu-
lating the crop responses to globally uniform environmental perturbations, including
CO2, temperature, precipitation, nitrogen, and adaptation (CTWN-A). The simulation
protocols are described in detail and key model outputs are made publically available.
The authors made the first cut on data analysis to show the key characteristics of the
simulated dataset. Overall, this manuscript is well organized and written. It also fulfills
the scope of GMD and should be of great interests to the broader crop modeling and
climate change adaptation community. I have the following comments for the authors

to consider:

Firstly, I see the nitrogen application rates designed in Table 1 are largely not realistic, especially considering how nitrogen application rates differ for different crops. I am not sure if I misunderstood anything there, but please help to clarify this point.

Secondly, I found some critical information is missing in the current manuscript. For example, the differences among different models (especially those with the same base), the irrigation triggering rules in different models, key model inputs (such as cultivar information), model tuning method and model spin-up design. Please see later detailed comments.

Thirdly, there are 7 mandatory variables in Table 2. However, the authors only discussed yield, which I agree is the most important one for crop models. If the authors can have some discussions on other variables, it would be very interesting, even if the figures are dumped into supplementary materials.

More detailed comments are as following:

P2, L30-L31: the transition to "Global crop model experiments are needed for systematic climate change assessments" is a little wired to me. Are you talking about the same point with last sentence or not?

P3, L22: Folberth et al. (2016); Porwollik et al. (2017)-> Folberth et al. (2016) and Porwollik et al. (2017)?

P3, L25: (C3MP Ruane et al., 2014; McDermid et al., 2015)-> (C3MP) (Ruane et al., 2014; McDermid et al., 2015)

P4, L26: an additional 84 for irrigated ($W\infty$)->an additional 84 for irrigated area ($W\infty$)? Are those 84 cases for irrigated area only with the assumption that the irrigated area will not change or also for rainfed area too (to get rid of water stress in rainfed regions)? Please clarify this point. It would be really interesting to have a no-water-limitation case for rainfed area. Moreover, how does each model trigger irrigation? Does the irrigated

[Figure]

amount differ a lot among models?

Table 1: There are three levels of applied nitrogen (10, 60, 200 kg/ha). Are those three levels uniformly applied for all the five crops? For soybean, we don't need that much nitrogen (200 kg/ha), right? For corn, is 10 kg/ha a too strong nitrogen limitation, especially for a few regions such as US?

P7, L5: it would be good to document the main differences related with crop growth among those sharing-a-common-base models, i.e. EPIC group (EPIC-IIASA, EPIC-TAMU, GEPIC, pEPIC), and LPJ group (LPJml, LPJ-GUESS).

P7, L24-L25: will the change of phenological parameters have a huge impact on yield for different models?

P7, L28: what's the "technical reasons" for CARAIB model? A note should be put on this.

P7L35-P8L1: how did modelers adjust those parameters? Was it manual tuning or automatic tuning? And should this tuning be conducted for every year and each location? Ideally, there should be a section in the appendix for parameter tuning to include related details (parameter space, and tuning method)

P9, L10: please move "(Note that several models do not output the anthesis date.)" after "the dates of planting, anthesis, and maturity", i.e. the dates of planting, anthesis, and maturity (Note that several models do not output the anthesis date).

P9, L8: 30-year or 31-year (1980-2010)? What's the model spin-up protocol?

P11, L20: no italic text in Table 3!

P11, L28: did you missed 510 ppm there?

P13, L26: For example, global correlation coefficients for maize in Phase I and Phase II are 0.89 and 0.74, respectively; for wheat 0.67 and 0.64, and for soybeans 0.64 and 0.59. (Compare to Müller et al. (2017) Figures 1–4 and 6.)-> For example, global

correlation coefficients in Phase I and Phase II are 0.89 and 0.74 for maize, 0.67 and 0.64 for wheat, and 0.64 and 0.59 for soybeans, respectively (Phase I values are from Figures 1-4 and 6 in Müller et al. (2017))

P13, L27: Figure 2 should be Figure 2(c)-2(f)

Caption of Figure 5: There are two "all" in "Figure shows all all simulated grid cells for each model"

P19, L1: region. (For soybeans, temperature effects are more complex; see Supplemental Figure S5.)-> region (for soybeans, temperature effects are more complex; see Supplemental Figure S5).

P20, L10: Generally, the carbon fertilization effect (CFE) would be larger under drier condition than under wetter condition. Is this true in Fig. 6a and Fig. S7?? McGrath, J.M., & Lobell, D.B. (2013). Regional disparities in the CO2 fertilization effect and implications for crop yields. Environmental Research Letters, 8, 014054

P21, L1-2: again, please check the use of parenthesis.

Section 5: I am glad that the authors discussed some of the limitations in the simulation exercise. One more point should be included there is about how to validate the simulated responses, especially considering that there are indeed some field experiments designed to measure the responses of crops to environmental manipulations.

---

## Author Response (AR1)

**Author resonse to referee comments on "The GGCMI Phase 2 experiment: global gridded crop model simulations under uniform changes in CO$_2$, temperature, water, and nitrogen levels (protocol version 1.0)"**

James Franke et al.[1,2]

[1]Department of the Geophysical Sciences, University of Chicago, Chicago, IL, USA
[2]Center for Robust Decision-making on Climate and Energy Policy (RDCEP), University of Chicago, Chicago, IL, USA

**Correspondence:** James Franke (jfranke@uchicago.edu)

**1   Cover letter to the Editor**

March 13, 2020

Dear Editor,

Attached is the author response to referee comments and the modified manuscript. The largest change involves Subsection 2.2, which has been substantially modified to address reviewer comments. The section now more clearly outlines the rationale for the uniform offsets used in the experiment design. This was the main concern raised by both reviewers. Other significant modifications to address reviewer comments include: a table detailing model differences has been added to the supplement (Table S1), and three figures have been added to the supplement that show model responses to other dimensions as requested by a reviewer. Because the original manuscript led both reviewers to miss key elements of the underlying motivation for the experiment, we have also made numerous minor adjustments throughout the paper to clarify the writing. Please note that we changed "Phase II" throughout the paper to "Phase 2" to match previous publication convention. This includes in the title of the paper. No figures have been modified. Attached are:

– The response to the referee comments, with explanation of line-by-line changes to the manuscript. *Original comments in gray* and author responses in black.

– The pdf with all the text modifications highlighted (using difflatex). Red text has been removed and blue text has been added.

Thank you,
James Franke (and coauthors)
University of Chicago

**2    Anonymous Referee 1**

*COMMENT: The manuscript by Franke et al., details the experimental design for the Phase II GGCMI crop model comparison. The goal is to provide a set of simulations to synchronize a variety of crop models and compare the responses from perturbations of temperature, precipitation, CO2, and nitrogen fertilizer. The result is a dataset of thousands of simulations that can be used to emulate statistical crop model response under varying inputs of climate change. The authors provide some analysis of the dataset, providing examples of non-linear behavior under multiple variable perturbations be-tween temperature, precipitation, and CO2. Furthermore, the authors provide access to other users for additional analysis. The manuscript is well written, the message is clearly defined, with a logical flow throughout, and void of technical errors. The authors did a good job detailing some of the more complex features of their study.*

RESPONSE: Thank you for the overall assessment.

*COMMENT: My main concern with this manuscript is I find the approach toward the perturbation experiments somewhat unrealistic. I understand the difficulty in generating simulations across different models in a way that is uniform, and I find the large number of simulations included in the dataset impressive but having such a large set of parameters for the simulations makes interpreting the output difficult and negates the heterogeneous(in space and time) behavior of climate. Wouldn't it be easier to use CMIP output to drive simulations which could reduce the number of model runs? Perhaps the authors could provide more discussion on this choice. Also, since these are offline runs, they don't include feedbacks between the atmosphere and land (e.g. irrigation feedbacks to temperature), which are important.*

RESPONSE: Yes, the approach of using uniformly perturbed climate inputs does not reflect realistic climate scenarios. If the goal of GGCMI Phase 2 were to use these simulations for climate change impact assessments, this experimental design would be the wrong choice. However, the goals of GGCMI-2 are to (i) scrutinize model response in response to individual and combined drivers and to (ii) develop crop model emulators on these experiments. Both of those goals require sampling across the space of potential perturbations that allows untangling the contributions of individual factors that are highly correlated in realistic future scenarios (e.g. CO2 and temperature). That is, meeting our goals requires a suite of unrealistic inputs.

We do believe that GGCMI-2 can also serve the needs of impacts assessments through the development of emulators. That is, the responses to the CTWN-A factors diagnosed from GGCMI-2 can be used to build up emulations of what the crop models would produce under a realistic climate scenario, including all the heterogeneous aspects of true climate change. (In this exercise, a crop model emulator for each individual grid cell is driven with the timeseries of projected climate changes for that particular location.)

This use is not demonstrated in the manuscript here, which is the "experiment description" paper; instead it is shown in a companion "model description" paper that describes the CTWN-A emulators, and is now available as a GMD discussion paper (https://www.geosci-model-dev-discuss.net/gmd-2019-365/ ). GMD had requested that we split our discussion of GGCMI-2 into these two components, to clearly distinguish the experimental description from the emulator development. In the companion paper we show that realistic CMIP-based simulations can be reproduced extremely well by emulators built from the CTWN-A experiment. This is true despite the fact that the uniform offset experiments omit some aspects of climate change

that could be important to crops - the distribution of weather conditions within growing season, e.g. stronger warming in spring vs. summer. Such effects do not appear large enough to compromise the GGCMI-2 emulators.

We have revised the text of the manuscript under review here to better explain the rationale of GGCMI-2 and to point to the companion emulator paper as a justification of its utility for impacts assessment. This point is extremely important and we thank the reviewer for pointing out that our explanation was insufficiently clear in the submitted manuscript.

It is true that these experiments do not include feedbacks between irrigation and temperature. This is non-ideal, but in practice, the vast majority of crop yield projections under climate change also omit this feedback, and instead simply feed a climate projection to a process-based or statistical crop model. We have added some discussion of this point in the manuscript and suggested the need for future studies.

CHANGES:

- Extensive justification for the experiment set-up has been added to pages 6 and 7.

- The companion paper which details emulation for impact assessment is now cited on page 7, line 19.

- Feedbacks between atmospheres and temperatures are now addressed in new sentences on page 24, lines 1 through 3.

*COMMENT: I did not find the A1 simulations discussed anywhere. They seem to be included in the methods section but are not included in the analysis. Perhaps they should be omitted. Similarly, the nitrogen simulations are also missing from the analysis (except for the correlation with observations).*

RESPONSE: The A dimension (adaptation in growing season length) is an integral part of the protocol and should be described fully in the experiment description paper. We have now given it more attention in the overall paper.

Note that the adaptation dimension will always be treated somewhat separately in discussion as it is not directly comparable to the other four dimensions (CTWN): it does not address inputs but the parameterization of crop varieties.

We have tried to limit the amount of analysis of results that are shown as experiment description papers are supposed to focus on experimental design, with a few results only as illustrations. However, we have added some additional material on N and A so that these dimensions do not seem less important.

CHANGES:

- A figure has been added to the supplement (S14) that shows the yield response to increased temperature in the A1 scenarios.

- Some discussion of the difference between A1 and A0 response has been added to page 18 line 21 - 25. Clarification between which scenario is being referenced in each case has be added throughout.

- New discussion of model calibration procedures for A1 growing seasons has been added to the supplement page 5.

- A figure has been added to the supplement (S15) that shows the yield response to increased temperature across the nitrogen dimension.

*General Comments:*

*COMMENT: P. 7, Section 2.3: The 12 models included in the study are very different types of models. I know this was discussed in the original paper describing protocol I, but it should also be noted here. How did the model differences inform the experimental design (or limit the scope of the study)?*

RESPONSE: Yes, the inclusion of different model types in a model intercomparison both complicates and enriches the analysis. One goal of the GGCMI Phase 2 experiment is to analyze model differences in order to better understand skills and deficiencies and to improve models. We have added this point more clearly in the text. Based on reviewer comments, we are also now adding a section describing key differences among models, and including a table in the supplemental material that describes model differences in inputs, structure and setup.

CHANGES:

– A new paragraph describing model differences has been added to page 12, lines 25-33.

– A new table has been added to the supplement (Table S1) detailing model differences.

*COMMENT: P. 9, L. 10: If some models don't output the anthesis date, why is it considered mandatory?*

RESPONSE: The anthesis date is an important phenological indicator and was considered a standard output also in the previous stages of GGCMI. However, as some models do not explicitly compute anthesis dates, these cannot deliver these outputs. The "mandatory" label means that models that do compute anthesis should report it. We have modified the text and table caption to make this clear.

CHANGES:

– Table caption modified as noted on page 11.

– The "mandatory" designation has been clarified on page 10, line 26.

*COMMENT: P. 15, L. 6: Is the negative impact on yield from increasing temperature due to shorter growing seasons or from actual heat damage to the crop?*

RESPONSE: Typically, the effect is a combination of the two mechanisms. The use of both A0 and A1 setups was designed to answer exactly this question. We have added text to emphasize this point. Note however that the experiment description paper here is not intended to conduct all these analyses, but rather to describe the protocol and outputs of the experiments that will allow questions to be answered. We are glad that the experiment provokes such useful responses! GGCMI team members are currently preparing a paper describing in detail the effects of adaptation in these experiments.

CHANGES:

– Text added as noted on page 17, line 15.

*COMMENT: P. 15, L 11-13: The change in yields at different latitudes is unrealistic because of the design of the experiment. Simply increasing temperature uniformly and not accounting for the seasonal differences in temperature change (i.e., stronger*

*winter increase in temperature and weak or no summer increase) results in an unrealistic "warming" during the growing season that might not exist. This is also the probable cause of the increase in yield from the least realistic simulations (Pl. 15, L 28-29).*

RESPONSE: As discussed above, the uniform perturbations are not intended to reproduce a realistic scenario and should not be used as such. The reviewer's comment is useful in telling us that we need to make this point more clear in the paper. We have added language to emphasize that the GGCMI-2 output for a given uniform temperature shift should not be taken as a proxy for an actual projection under a realistic climate scenario that produces the equivalent global mean temperature change.

We have tried to clarify two important points brought up by this comment. First, the climate offsets in the GGCMI-2 experiments refer to offsets during the growing season, not to annual means. The strong increases in yield in high-latitude regions in some simulations are therefore the appropriate response for each model given the applied level of warming during the growing season. Models of course disagree on the extent or even the sign of yield changes, especially in high-latitude regions, and their responses may be unrealistic.

Second, a scenario with a uniform offset (across space and time) will not match a scenario with the same mean change but with the spatial patterns of climate change expected under future scenarios. The effects of these spatial patterns are shown explicitly in the companion GMD "model description" paper (see link above). We now refer to that paper explicitly.

CHANGES:

– New paragraph added as noted on page 7, lines 7-20 as noted.

**3 Anonymous Referee 2**

*COMMENT: The manuscript by Franke et al. documented a new AgMIP GGCMI effort on simulating the crop responses to globally uniform environmental perturbations, includingCO2, temperature, precipitation, nitrogen, and adaptation (CTWN-A). The simulation protocols are described in detail and key model outputs are made publically available. The authors made the first cut on data analysis to show the key characteristics of the simulated dataset. Overall, this manuscript is well organized and written. It also fulfills the scope of GMD and should be of great interests to the broader crop modeling and climate change adaptation community.*

RESPONSE: Thank you for the overall positive assessment.

*COMMENT: I have the following comments for the authors to consider:*

*COMMENT: Firstly, I see the nitrogen application rates designed in Table 1 are largely not realistic, especially considering how nitrogen application rates differ for different crops. I am not sure if I misunderstood anything there, but please help to clarify this point.*

RESPONSE: The idea of the uniform perturbation and input levels in the CTWN-A experiment is not to be fully realistic but to allow for in-depth analyses by providing a structured analysis framework. Fertilizer application rates differ substantially across crops (e.g. maize vs. soybean) but also across the globe where access to fertilizers is often limited. We designed the ranges of CTWN so that low and high-end values are included and model behavior can be understood across these dimensions.

By using a range of nitrogen input levels (as well as inputs of climate variables), we are able to construct "emulators" of the crop model responses to these factors for arbitrary input levels. That is, the GGCMI-2 experiments allow constructing a response surface that would allow reproducing the output crop models would have produced if run with more realistic (or indeed, any) nitrogen inputs. This use is explained in detail in a companion GCM paper now available online at (https://www.geosci-model-dev-discuss.net/gmd-2019-365/ ). We have added more discussion in this first "experiment description" paper to make this clear, and now point to the companion paper.

CHANGES:

– Text added on page 6, line 11 to clarify the intent of the N application levels.

*COMMENT: Secondly, I found some critical information is missing in the current manuscript. For example, the differences among different models (especially those with the same base),the irrigation triggering rules in different models, key model inputs (such as cultivar in-formation), model tuning method and model spin-up design. Please see later detailed comments.*

RESPONSE: We agree with this critique and have expanded the discussion of structural differences among models. We have also included a table in the Supplemental Materials showing key model features and structural differences. This addition will make the manuscript significantly more useful for readers.

CHANGES:

– A new paragraph describing model differences has been added to page 12, lines 25-33.

– A new table has been added to the supplement (Table S1) detailing model differences.

*COMMENT: Thirdly, there are 7 mandatory variables in Table 2. However, the authors only discussed yield, which I agree is the most important one for crop models. If the authors can have some discussions on other variables, it would be very interesting, even if the figures are dumped into supplementary materials.*

RESPONSE: As this is the experiment description paper and the experiment is very comprehensive (experiments, different model types, different output variables), we tried to find a balance between producing a readable overview paper and one with exhaustive detail. However, we may have erred on the side of over-focusing on yield. We have therefore now added some examples of other outputs in the supplementary material and more discussion in the main text.

CHANGES:

– A figure illustrating the irrigation water response to warming has been added to the supplement (Figure S16).

*COMMENT: More detailed comments are as following: COMMENT: P2, L30-L31: the transition to "Global crop model experiments are needed for systematic climate change assessments" is a little wired to me. Are you talking about the same point with last sentence or not?*

RESPONSE: Yes, this sentence is a bit too condensed. We have updated the language accordingly.

CHANGES:

– Text modified on page 2, line 27-33 to clarify meaning.

*COMMENT: P3, L22: Folberth et al. (2016); Porwollik et al. (2017)-> Folberth et al. (2016) and Porwollik et al. (2017)?*

RESPONSE: Thank you for catching this! These references are now corrected; we have updated Folberth et al. 2016 to Folberth et al. 2019.

CHANGES:

– Correction made on page 3, line 28.

*COMMENT: P3, L25: (C3MP Ruane et al., 2014; McDermid et al., 2015)-> (C3MP) (Ruane et al.,2014; McDermid et al., 2015)*

RESPONSE: We have changed to "(C3MP, see Ruane et al., 2014; McDermid et al., 2015)"

CHANGES:

– Correction made on page 3, line 33.

*COMMENT: P4, L26: an additional 84 for irrigated (W∞)->an additional 84 for irrigated area (W∞)?Are those 84 cases for irrigated area only with the assumption that the irrigated area will not change or also for rainfed area too (to get rid of water stress in rainfed regions)? Please clarify this point. It would be really interesting to have a no-water-limitation case for rainfed area. Moreover, how does each model trigger irrigation? Does the irrigated amount differ a lot among models?*

RESPONSE: No, following the general GGCMI experiment design, fully rainfed and fully irrigated systems are simulated in all grid cells, independent of their actual distribution. This protocol allows for better analyses (e.g. simulations with and

without water stress can be compared) and also for understanding the potential of optional cropland expansion. In the GGCMI Phase 2 setup, irrigated systems are also simulated during the rainfed growing seasons so that the simulation results are directly comparable and only differ with respect to water supply. We have modified the text to make this choice more clear.

CHANGES:

5   – Text added on page 6, lines 1-6 line 3 to clarify the irrigation protocol.

   – Table added to supplement detailing individual model protocol for irrigation triggering (Table S1).

*COMMENT: Table 1: There are three levels of applied nitrogen (10, 60, 200 kg/ha). Are those three levels uniformly applied for all the five crops? For soybean, we don't need that much nitrogen (200 kg/ha), right? For corn, is 10 kg/ha a too strong nitrogen limitation, especially for a few regions such as US?*

10  RESPONSE: Yes, as discussed above, N levels in the experimental protocol are uniformly applied across all locations and all crops, and are not intended to be realistic. The experiment design is intended to span the full range of plausible input values, though the maximum of 200 kgN/ha may actually be a bit low for some crops and regions. Using uniform offsets and input levels allows structured analysis of the effects of each factor. As per answers above, we have now made this rationale more clear in the text.

15  CHANGES:

   – Text added on page 6, lines 11 to clarify the intent of the N application levels.

*COMMENT: P7, L5: it would be good to document the main differences related with crop growth among those sharing-a-common-base models, i.e. EPIC group (EPIC-IIASA, EPIC-TAMU, GEPIC, pEPIC), and LPJ group (LPJml, LPJ-GUESS).*

RESPONSE: Yes, as discussed above, these differences are included in the table with details on model inputs, structure and
20  setup that is now included in Supplementary Material. We feel this addition greatly strengthens the utility of the paper and thank the reviewer for the suggestion.

CHANGES:

   – A new paragraph describing model differences has been added to page 12, lines 25-33 including some discussion about models that share a common genealogy.

25  – A new table has been added to the supplement (Table S1) detailing model differences.

*COMMENT: P7, L24-L25: will the change of phenological parameters have a huge impact on yield for different models?*

RESPONSE: Yes, model performance can be very sensitive to the parametrization of growing seasons. That is why the experiment protocol prescribes harmonized growing seasons so that it is easier to analyze model responses. We have amplified discussion of this point in the text.

30  CHANGES:

– Sentence added to page 8, line 28.

*COMMENT: P7, L28: what's the "technical reasons" for CARAIB model? A note should be put on this.*

RESPONSE: We agree that "technical reasons" does not adequately describe the issue at hand. In fact, the CARAIB team simply missed harmonizing this aspect. We think it is still of value to include their output in the archive, and any applications can exclude CARAIB results if required for their purposes. We have adjusted the sentence in question.

CHANGES:

– Text modified in the paragraph starting on page 8, line 24 to clarify the process here.

*COMMENT: P7L35-P8L1: how did modelers adjust those parameters? Was it manual tuning or automatic tuning? And should this tuning be conducted for every year and each location? Ideally, there should be a section in the appendix for parameter tuning to include related details (parameter space, and tuning method)*

RESPONSE: We now describe this procedure more clearly in the main text. The groups used manual parameter tuning to harmonize the growing seasons. First, parameters are adjusted for each crop in each location under the unperturbed AgMERRA baseline climate timeseries so that growing seasons in this 31-year period (1980-2010) reproduce specified observed average growing seasons for this period. For A0 simulations, the parameters are then left constant for all experiments, so that growing seasons alter under warming.

Note that because each crop model sets the growing season differently, the parameters modified will differ across models. Describing the exact procedure of the different modeling groups would require extensive discussion of the structure of each model. While we agree an appendix describing this would be useful to some readers, we feel it is out of scope for this paper. We hope that this need is satisfied instead by our links to the description papers for each individual model, which should cover their process of determining growing seasons.

CHANGES:

– Text modified in the paragraph on page 9, lines 8-22 to better describe the growing season calibration.

– New list of A1 case calibration measures added to the supplement on page 5.

*COMMENT: P9, L10: please move "(Note that several models do not output the anthesis date.)"after "the dates of planting, anthesis, and maturity", i.e. the dates of planting, anthesis, and maturity (Note that several models do not output the anthesis date).*

RESPONSE: Reviewer 1 also had problems with this sentence and it has been revised accordingly. It no longer includes parentheses.

CHANGES:

– Correction made on page 10, line 26.

*COMMENT: P9, L8: 30-year or 31-year (1980-2010)? What's the model spin-up protocol?*

RESPONSE: The spin up is very different across models. This is now documented in the new table on model inputs, structure and setup.

CHANGES:

– New table added to supplement (Table S1).

*COMMENT: P11, L20: no italic text in Table 3!*

RESPONSE: Thank you for catching this. These simulation sets are shown in bold in Table 3 (column "Sims per crop"), and the sentence is now corrected.

CHANGES:

– Correction made on page 14, line 13.

*COMMENT: P11, L28: did you missed 510 ppm there?*

RESPONSE: Yes, thanks for catching this; it is now corrected.

CHANGES:

– Correction made on page 14, line 21.

*COMMENT: P13, L26: For example, global correlation coefficients for maize in Phase I and Phase II are 0.89 and 0.74, respectively; for wheat 0.67 and 0.64, and for soybeans 0.64 and 0.59. (Compare to Müller et al. (2017) Figures 1–4 and 6.)-> For example, global correlation coefficients in Phase I and Phase II are 0.89 and 0.74 for maize, 0.67 and0.64 for wheat, and 0.64 and 0.59 for soybeans, respectively (Phase I values are from Figures 1-4 and 6 in Müller et al. (2017))*

RESPONSE: Corrected as suggested.

CHANGES:

– Correction made on page 15, line 28.

*COMMENT: P13, L27: Figure 2 should be Figure 2(c)-2(f)*

RESPONSE: Corrected as suggested.

CHANGES:

– Correction made on page 15, line 30.

*COMMENT: Caption of Figure 5: There are two "all" in "Figure shows all all simulated grid cells for each model"*

RESPONSE: Thanks for catching this; corrected.

CHANGES:

– Correction made to Figure 5 caption on page 20.

*COMMENT: P19, L1: region. (For soybeans, temperature effects are more complex; see Supple-mental Figure S5.)-> region (for soybeans, temperature effects are more complex; see Supplemental Figure S5).*

RESPONSE: Corrected as suggested.

CHANGES:

– Correction made on page 22, line 2.

*COMMENT: P20, L10: Generally, the carbon fertilization effect (CFE) would be larger under drier condition than under wetter condition. Is this true in Fig. 6a and Fig. S7? McGrath, J.M., & Lobell, D.B. (2013). Regional disparities in the CO2 fertilization effect and implications for crop yields. Environmental Research Letters, 8, 014054*

RESPONSE: The GGCMI-2 experiment is designed to allow diagnosis of this and other interaction effects! But, as this is the experiment description paper, we are not analyzing results in full depth. We hope instead that many analyses will follow, making use of the freely available data set that we describe here. We have now added this citation and mentioned this effect as the possible target of a future study.

CHANGES:

– Citation adde to page 22, line 23.

– Text added to page 24, line 8.

*COMMENT: P21, L1-2: again, please check the use of parenthesis.*

RESPONSE: We have removed parentheses here.

CHANGES:

– Correction made on page 23, line 11. Text modifications for clarity.

*COMMENT: Section 5: I am glad that the authors discussed some of the limitations in the simulation exercise. One more point should be included there is about how to validate the simulated responses, especially considering that there are indeed some field experiments designed to measure the responses of crops to environmental manipulations.*

RESPONSE: We now include more discussion of the fact that models have been individually and jointly evaluated, including against data from field experiments.

We also discuss the challenges from the artificial model setup in the GGCMI Phase 2 experiment more thoroughly, and now refer to the companion paper (Franke et al., 2020), in which we demonstrate that emulators built from this artificial setup can very well reproduce model behavior from crop yield simulations driven by more realistic future climate projections.

CHANGES:

– Text added to address model validation on page 15, lines 6-9.

– Text added to address model validation against realistic climate scenarios with the emulator on page 23, line 31.

Reference:

[revised manuscript text omitted]